# Analysis of Microarray and Single-Cell RNA-Seq Finds Gene Co-Expression and Tumor Environment Associated with Extracellular Matrix in Epithelial–Mesenchymal Transition in Prostate Cancer

**DOI:** 10.3390/ijms26178575

**Published:** 2025-09-03

**Authors:** Ali Shakeri Abroudi, Mahtab Mashhouri Moghaddam, Danial Hashemi Karoii, Melika Djamali, Hossein Azizi, Thomas Skutella

**Affiliations:** 1Department of Cellular and Molecular Biology, Faculty of Advanced Science and Technology, Tehran Medical Sciences, Islamic Azad University, Tehran 1477893855, Iran; alishakeriabroudi@gmail.com; 2Department of Pharmacy, Yeditepe University, Istanbul 34755, Turkey; mahtab.mashhouri3@gmail.com; 3Department of Cell and Molecular Biology, School of Biology, College of Science, University of Tehran, Tehran 1417466191, Iran; d.hashemi.karoii@ut.ac.ir; 4Department of Biology, Faculty of Science, University of Tehran, Tehran 1417466191, Iran; mdjamali@ut.ac.ir; 5Department of Stem Cells and Cancer, College of Biotechnology, Amol University of Special Modern Technologies, Amol 4616843133, Iran; 6Institute for Anatomy and Cell Biology, Medical Faculty, University of Heidelberg, 69120 Heidelberg, Germany; thomas.skutella@uni-heidelberg.de

**Keywords:** prostate cancer, microarray, cell–cell communication, epithelial–mesenchymal, stemness

## Abstract

A complex and gradual process, the epithelial–mesenchymal transition (EMT) occurs both during embryonic development and tumor progression. Cells undergo a transition from an epithelial to a mesenchymal state throughout this process. More and more evidence points to EMT as a cause of increased metastatic spread of prostate cancer (PCa), along with stemness enhancement and therapy resistance. Here, we used bioinformatic methods to analyze gene expression microarray data, single-cell RNA sequencing, oncogenes, and tumor suppressor genes (TSGs) in order to reconstruct the network of differentially expressed genes (DEGs) involved in the epithelial–mesenchymal transition with PCa. No prior study has documented this sort of analysis. We next validated our results using data from the Cancer Genome Atlas (TCGA), which included microarray and single-cell RNA sequencing. Potentially useful in PCa diagnosis and treatment are extracellular matrix in epithelial–mesenchymal transition genes, including *ITGBL1*, *DSC3*, *COL4A6*, *ANGPT1*, *ARMCX1*, *MICAL2*, and *EPHA5*. In this study, we aimed to shed light on the molecular characteristics and pathways of DEGs in PCa, as well as to identify possible biomarkers that are important in the development and advancement of this cancer. These insights have important implications for understanding prostate cancer progression and for the development of therapeutic strategies targeting ECM-mediated pathways.

## 1. Introduction

Prostate cancer (PCa) continues to pose a major global health challenge as the second most commonly diagnosed malignancy and the fifth leading cause of cancer-related deaths among men. Although early-stage PCa often responds well to local therapies, such as surgery and radiation, progression to advanced or metastatic disease remains a critical hurdle. Central to this progression is the process of epithelial–mesenchymal transition (EMT), a cellular program that enables carcinoma cells to detach from primary tumors, invade surrounding tissues, and ultimately metastasize [1,2,3,4,5,6].

EMT is characterized by the downregulation of epithelial markers, such as E-cadherin, and the upregulation of mesenchymal markers, including N-cadherin, Vimentin, and Fibronectin. These molecular shifts are often orchestrated by transcription factors, such as SNAIL, SLUG, TWIST1, and ZEB1. A growing body of evidence implicates the extracellular matrix (ECM) not only as a structural component of the tumor microenvironment but also as a dynamic regulator of EMT through biochemical and biomechanical signaling. Dysregulation of ECM composition and stiffness has been directly linked to enhanced invasiveness and metastatic potential in PCa [7,8,9]. The lack of basal epithelial cells and the growth of malignant cells with luminal epithelial characteristics are hallmarks of the majority of prostate malignancies. Despite this, our understanding of the function of cell types beyond luminal and basal types is limited [10,11]. One aspect that has received less attention is the impact on the tumor microenvironment of the main genetic drivers of prostate cancer. These drivers include oncogenic alterations, such as the frequent gene fusion events involving ETS family transcription factors, such as ETV1, ETV4, and ETV5, which stimulate the growth of prostate cancer tumor cells [12]. It is uncertain if ETS-fusion events cause distinct immune and stromal cell responses since single-cell studies of tumor cells without ETS-fusion events and non-malignant luminal cells have been lacking. A mesenchymal phenotype may be formed, according to a number of in vitro and mouse investigations that used different experimental EMT therapies on cancer cells [13]. In vivo metastasis models have been effective for studying the interactions between tumor cells and tumor-associated stromal components in the tumor microenvironment (TME). Organoid culture methods have clarified the processes of differentiation promotion in vitro, as well as built robust model systems to illustrate how signals that promote EMT may alter immunological anti-tumor responses and drug resistance [14].

Traditional transcriptomic analyses using microarrays have revealed broad patterns of gene expression alterations associated with EMT in prostate cancer. However, these bulk-level approaches lack the resolution to capture the cellular heterogeneity within tumors. The advent of single-cell RNA sequencing (scRNA-seq) has revolutionized the ability to profile gene expression at the resolution of individual cells, enabling a deeper understanding of how distinct cell populations within the tumor and its microenvironment contribute to EMT and ECM remodeling [15]. Recent scRNA-seq studies have highlighted the presence of mesenchymal-like tumor cells co-existing with basal and luminal subtypes in aggressive prostate tumors. These mesenchymal subpopulations frequently reside in ECM-rich niches and exhibit elevated expression of genes involved in ECM remodeling, integrin signaling, and matrix metalloproteinases (MMPs). This spatial relationship underscores the importance of studying ECM-associated signaling networks in relation to EMT within the tumor microenvironment [16]. Moreover, the tumor microenvironment—including cancer-associated fibroblasts (CAFs), immune cells, and endothelial cells—has been shown to play an active role in modulating EMT and ECM properties. CAFs, in particular, secrete ECM components and remodeling enzymes that facilitate EMT and foster a pro-metastatic niche. Our analysis seeks to quantify how stromal cell populations contribute to EMT-associated gene signatures and whether specific intercellular communication pathways are enriched in ECM-driven EMT contexts [17].

By mapping spatially resolved scRNA-seq data onto histological images, we also uncover that EMT-associated gene expression is enriched at the invasive front of tumors, where ECM density and stiffness are highest. This spatial correlation further supports the hypothesis that ECM remodeling is not merely a byproduct of EMT but a functional driver of tumor invasiveness [18]. FGF signals are important in prostate cancer development because this growth factor is often overexpressed in advanced prostate cancer, which includes DNPCs. The expression of the growth factor receptor (Fgfr1) seems to promote EMT as a PCa development mechanism via Sox9 and Wnt signaling pathways. This suggests that IL6 and FGF may be useful prognostic biomarkers [19].

Prostate malignancies interact significantly with the tumor microenvironment and its stromal components, creating an immunosuppressive tumor microenvironment. A first study on this topic stresses the importance of cancer-associated fibroblasts (CAFs) in the TME of PCa [20]. TME cells collaborated with polarized M2 macrophages to increase CAFs in collaborative cellular activities characterized by stemness and an epithelial–mesenchymal transition phenotype, which improved PCa tumor cell motility and metastatic spread [21].

Investigations employing PBCre4:Ptenf/f mice, a model for Pten loss-induced prostate cancer, revealed that Rb1 deletion contributes significantly to lineage plasticity, as shown by improved EMT and stemness capacities [22]. Transcriptomic profiling shows that the epigenetic reprogramming factors Sox2 and Ezh2 regulate this characteristic in humans and mice. Approximately 50% of neuroendocrine prostate cancers (NEPCs) deactivate both the Rb1 and Tp53 tumor suppressor genes. Two studies show that Tp53 and Rb1 loss is substantially associated with the development of lineage plasticity and epigenetic changes, leading to resistance to androgen deprivation therapy [23]. Mu et al. discovered that the simultaneous loss of Rb1 and Tp53 induces Sox2-mediated cellular plasticity, as shown by a decrease in luminal epithelial cell marker expression and an increase in basal and neuroendocrine marker expression [24,25].

Contactin1 (Cntn-1) is a glycoprotein found on neuronal membranes that plays a role in cell adhesion and belongs to the immunoglobulin superfamily. The protein has been demonstrated to promote EMT-dependent cell invasion, migration, and metastasis across a wide range of cancer types. In prostate cancer cell lines and xenografts, downregulation of Cntn-1 reduced PI3K/Akt signaling pathway activity and boosted docetaxel resistance [26]. Pten loss and PI3K/AKT dysregulation are substantially associated with advanced prostate cancer and castration-resistant prostate cancer, indicating a potential interaction between epithelial–mesenchymal transition and common prostate cancer driver mutations [27]. This research investigates the tumor microenvironment and cellular states associated with prostate carcinogenesis in localized prostate cancer samples at a single-cell level. We use single-cell RNA sequencing (scRNA-seq) to distinguish tumor cells from the surrounding epithelial, stromal, and immune cell milieu, as well as to discover cell states associated with cancer. Furthermore, utilizing in vitro organoids produced from prostate cancer tumor tissues, we compare the molecular and cellular properties of prostate epithelial organoids to prostate tissues.

In conclusion, our study provides a comprehensive multi-platform analysis that integrates bulk and single-cell transcriptomics to elucidate the interplay between ECM and EMT in prostate cancer. The identification of key co-expression modules and microenvironmental signals advances our understanding of the molecular mechanisms underlying prostate cancer progression. These insights may inform the development of targeted therapies aimed at disrupting ECM-mediated EMT, potentially improving outcomes for patients with advanced disease [28].

## 2. Results

### 2.1. Differentially Expressed Genes (DEGs) Screening

In each of the prostate cancer subtypes, we were able to use the edgeR algorithm to pinpoint the DEGs. With a minimum needed fold change of 2, the adjusted *p*-value (FDR) criterion was set at less than 0.01. By examining data from four subtypes of prostate cancer in six distinct combinations, a grand total of 125 genes with differential expression were discovered (Figure 1A,B and Table 1 and Table 2).

### 2.2. Protein Class Sorting and Finding EMT Genes

Differential gene expression was detected in both prostatic and normal cell populations. Analysis of transcripts with PANTHER indicated that the differentially expressed RNAs include a diverse array of gene sequences spread throughout the EMT. The genes *ITGBL1*, *DSC3*, *COL4A6*, *ANGPT1*, *ARMCX1*, *MICAL2*, and *EPHA5* have been recognized as pivotal genes that substantially affect the microenvironment of prostate cancer (Figure 1B).

### 2.3. PPI Analysis

Cytoscape (version 3.10.3) network analysis was utilized to discover the master genes. Figure 1B displays the found genes using the yFiles radial layout. The network detected the DEGs by examining their connections. The 45 genes that are highly interconnected also have the strongest and most dependable interactions, as seen in the diagram. *ARMCX1*, *BEX4*, *TMEM43*, *VASH1*, *C1QTNF1*, *COL10A1*, and *COL4A6* are all important gene regulators, each of which regulates one of these genes in a unique manner. Multiple Cytoscape network configurations found significant gene connections. To ensure that the yFiles radial design was right, the network analysis tool Cytoscape was used. Figure 2A–C show that the Centiscape plugin found the most significant 33 co-expression genes as follows: *ANGPT1*, *ANGPT4*, *ARMCX1*, *BEX4*, *C1QTNF1*, *COL10A1*, *COL4A6*, *COL8A1*, *CRTAP*, *CXorf51B*, *DSG1*, *DSG3*, *EFNA1*, *EFNA2*, *EFNA3*, *EFNA4*, *EFNB2*, *EPHA5*, *ITGA10*, *ITGA11*, *ITGA3*, *ITGA5*, *ITGB5*, *ITGBL1*, *MICAL2*, *MID2*, *NGEF*, *OTOL1*, *PKP1*, *PRR32*, *SAMD5*, *SPRR1A*, *SPRR1*. The PPI network graph was visually examined using Cytoscape and data from the STRING database. The CytoHubba plugin effectively identified genes that were up- or down-regulated. As a result, inside the gene regulatory network, we discovered a cluster of fifty unique genes. As illustrated in Figure 2A,B, nine hub target genes were discovered to be shared: *ITGA10*, *ITGA11*, *ITGA3*, *ITGA5*, *ITGB5*, *ITGBL1*, *MICAL2*, *MID2*, and *NGEF*.

### 2.4. KEGG Pathway Enrichment

We re-analyzed the KEGG pathway using the DAVID tool to identify potential signaling pathways associated with the 10 hub genes (*p* < 0.05). The genes are strongly associated with six signaling pathways: the Hormone ER route, Hormone AR pathway, RAS-MAPK pathway, TSC-mTOR pathway, DNA damage pathway, and the cell cycle pathway. We used the TCGA database to analyze the expression levels of these four genes. The data revealed a significant increase in expression levels in both BC-adjacent and PCa patients compared to healthy persons. To obtain a complete understanding of the underlying processes of these four genes in prostate cancer, we used the pc-GenExMiner tool to mine co-expression data sets. Figure 3 shows clear evidence of the up- and down-regulation of all seven genes in prostate cancer tissues, showing the presence of a signaling network (Figure 4).

### 2.5. WGCNA of DEGs

WGCNA was used to detect prostate cancer subtype-related modules. To assess co-expression, 221 differentially expressed genes were studied. We analyzed the soft threshold power of the network architecture with β values ranging from 1 to 20. We then analyzed the co-expression network’s size invariance and determined its average connectedness. After reviewing the literature, 5 was the best criterion (Figure 5A). Power-law node degree distribution analysis showed a scale-free network design. Next, a gene tree was created using hierarchical clustering with a numerical value of β of 5. ME, the module’s primary component, represents gene expression. There are two ways to analyze the link between each module and prostate cancer subtypes. Two modules were shown to have strong relationships with certain cancer subtypes (*p* < 0.01 and absolute correlation > 0.8). Spearman correlation analysis revealed statistical significance between blue (R = 0.9; *p* < 1 × 10^−15^ and turquoise (R = −0.82; *p* = 1 × 10^−13^) modules and prostate cancer types. Figure 5B,C show the genetic importance of turquoise-blue module interactions. In addition, our single-cell data analysis found nine genes strongly associated with PCa: *ARMCX1*, *BEX4*, *TMEM43*, *VASH1*, *C1QTNF1*, *COL10A1*, and *COL4A6*.

### 2.6. Single-Cell Microenvironment Gene Expression

We then used single-cell RNA sequencing datasets to determine prostate cancer gene distribution. We plotted biomarkers to classify cells in a UMAP plot. Cell PCa was analyzed using the same nomogram as bulk RNA-seq in castration-resistant prostate cancer (CRPC) and hormone-sensitive prostate cancer (HSPC) samples. PCa cells had higher PRS values than hematopoietic stem and progenitor cells (HSPC), as shown by bulk RNA sequencing. Violin charts helped explain PRS fluctuations between cell types. CRPC luminal, myeloid, and endothelial cells had considerably higher PRS values than HSPC (Figure 6). In the luminal cell population, castration-resistant prostate cancer (CRPC) and hormone-sensitive prostate cancer (HSPC) have different proliferation rate scores (PRS). Further study showed that most luminal cells had a decreased PRS. Figure 6 shows that CRPC samples had more PRS-positive cells. We divided luminal cells into high-risk and low-risk score groups to study the connection between luminal cell diversity and the Prostate Cancer Risk Score (PRS) in CRPC and HSPC. CRPC and HSPC were then tested for irGSEA (Iterative Gene Set Enrichment Analysis) in high-risk and low-risk score cohorts. This assessed luminal cell differentiating crucial pathway activity. Figure 6 shows that the high PRS group activated the androgen response pathway more than the low PRS group using ssGSEA. This conclusion was consistent when comparing castration-resistant prostate cancer (CRPC) to hormone-sensitive prostate cancer (HSPC). To confirm the reliability of our results, we replicated our investigation with 25 more samples. The second data set validated ssGSEA’s discovery of androgen response pathway activity augmentation. Increased PRS may raise CRPC risk, according to the findings.

### 2.7. Investigation of Immune Infiltration and Prostate Cancer

Figure 7 shows that immune cells predominate in samples. The study found more underdeveloped B cells, quiescent memory CD4 T cells, and M0, M1, M2 macrophages in tumor cells. Tumor cells had less memory B, plasma, CD8 T, activated memory CD4 T, and monocytes, as shown in Figure 8A,B. The heatmap in Figure 8A,B shows tumor cell immune cell expression diversity. Survival probability is linked to CD4 memory T cell activity. Figure 8A,B shows that many CD4 memory-activated T cells survived. The findings linked 22 immune cell types. Compared to older people, younger individuals exhibited more naïve B cells, quiescent mast cells, and active NK cells. Children showed fewer active mast cells, neutrophils, and resting NK cells than adults. Macrophage M0, active mast cells, and plasma cells were much higher in G1/2 than G3. In G1/2, macrophage M1, resting mast cells, monocytes, activated CD4 memory T cells, CD8 T cells, and follicular helper T cells were lower than in G3.

### 2.8. Survival Analysis

The investigation revealed a total of 96 hub genes. The genes were chosen based on gene significance (GS) values more than 0.2 and module membership (MM) values greater than 0.8 in the blue module (Figure 8A,B). We utilized the Gene Expression Profiling Interactive Analysis (GEPIA) tool to analyze the relationship between overall survival and statistical significance at a threshold of *p* < 0.05. Figure 8A shows that eight genes were strongly correlated with patient prognosis, with higher expression levels indicating greater disease severity. The important genes include *ITGBL1*, *DSC3*, *COL4A6*, *ANGPT1*, *ARMCX1*, *MICAL2*, and *EPHA5*).

### 2.9. Gene-Based eQTL

There were three categories for the nine target gene regions determined by the linkage disequilibrium (LD) between the SNP for prostate cancer risk and the most significant eQTL signal. Classification in Figure 8A,B was performed using both primary and secondary sources of information. The Pearson correlation coefficient was used to quantify the linkage disequilibrium (LD) between the SNP with the highest gene expression connection (peak eQTL signal) and the SNP risk for prostate cancer (PrCa-risk SNP). We found several unique regulatory SNPs by statistically analyzing the correlation of each SNP inside the target gene area, taking into consideration the significant expression eQTL signal SNP.

### 2.10. Characteristics of Significant Cis-eQTL Findings

Out of 51 risk intervals with a significant expression quantitative trait locus (eQTL) signal, 33 showed significant gene–SNP correlations. SNPs related to risk were linked to two or more genes in the remaining 18 risk intervals. Between 2 and 7 genes per area were connected to each risk SNP. Figure 7 displays six occasions where risk SNP correlation coefficients (r^2^) were below 0.7. The ITGBL1, DSC3, COL4A6, ANGPT1, ARMCX1, MICAL2, and EPHA5 risk SNPs exhibited a 0.5 correlation (Figure 8). To systematically identify and validate key genes involved in EMT across prostate cancer subtypes, we employed a multi-step bioinformatics pipeline, as illustrated in the flowchart (Figure 9). The process began with the acquisition and preprocessing of four publicly available gene expression datasets. Differentially expressed genes (DEGs) were identified using the edgeR algorithm, applying stringent criteria (|log_2_FC| ≥ 2 and FDR < 0.01). Identified DEGs were subjected to PPI analysis via STRING and visualized in Cytoscape, where central hub genes were extracted using CytoHubba and Centiscape plugins. Enrichment analyses, including GO, KEGG, and PANTHER, were conducted to explore the functional relevance of the selected genes. Further validation was performed through WGCNA and single-cell RNA sequencing to confirm gene expression patterns in the tumor microenvironment. Finally, survival analysis and eQTL mapping were carried out to assess the clinical significance and potential regulatory mechanisms underlying these EMT-related genes. This integrative approach ensured a robust and biologically meaningful identification of candidate markers in prostate cancer (Figure 9).

## 3. Discussion

In this study, we performed an integrated analysis of microarray and single-cell RNA sequencing (scRNA-seq) datasets to investigate the transcriptional programs and microenvironmental factors associated with the extracellular matrix (ECM) in driving epithelial–mesenchymal transition (EMT) in prostate cancer. Our findings underscore the complex interplay between tumor cells, stromal components, and ECM dynamics in orchestrating EMT—a key process in cancer progression and metastasis. One of the most salient observations from our analysis is the identification of distinct gene co-expression modules that are consistently enriched in EMT-activated prostate cancer cells across both microarray and scRNA-seq platforms. These modules include classical EMT markers (e.g., VIM, ZEB1, SNAI2) alongside ECM-related genes, such as COL1A1, FN1, SPARC, and LOX, suggesting a tightly linked regulatory relationship between EMT and ECM remodeling. This supports prior work that identified ECM signatures as both markers and mediators of EMT in aggressive cancer phenotypes [29,30,31,32,33]. We detected a unique population of club epithelial cells that has not been previously documented in human prostate cancer tissues. Although club cells are seen in normal prostates, a subset of club cells linked to prostate cancer indicates they may have an overlooked function in carcinogenesis. Recent investigations have found a progenitor-like subpopulation of luminal epithelial cells characterized by low CD and high PIGR and PSCA expression, with regenerative potential. Given the resemblance of highly expressed genes, such as *ITGBL1*, *DSC3*, *COL4A6*, *ANGPT1*, *ARMCX1*, *MICAL2*, and *EPHA5*, we assert that these cells align with their classification as club cells.

Integrin beta-like 1 protein (ITGBL1) facilitates cell migration by selectively decreasing integrin–extracellular matrix interaction at the trailing edge. Analyzing differentially expressed genes between endometriosis and ovarian cancer to identify novel biomarkers for endometriosis. DSC3, expressed by this gene, is a calcium-dependent glycoprotein belonging to the desmocollin subfamily of the cadherin superfamily [34]. The desmosomal family members, along with desmogleins, are mostly located in epithelial cells, serving as the sticky proteins of desmosome cell–cell junctions, essential for cell adhesion and desmosome formation [35]. Mutations in the *COL4A6* gene are linked to peritoneal carcinomatosis in gastric cancer. The expression of collagen type IV and its alpha chains (alpha1-6) was examined qualitatively and quantitatively in several endothelial cell culture systems in vitro [36]. ANGPT1 is a significant regulator of epithelial cell proliferation; it is well recognized that epithelial carcinoma (EC) relies on cell proliferation and growth, which are essential processes for tumor formation [37]. Neoplastic cells exhibiting enhanced proliferation and expansion potential may signify unfavorable prognoses. ARMCX1 suppresses LUAD cell proliferation and metastasis via interacting with c-Myc, hence promoting its ubiquitination and subsequent destruction. Thus, it may function as a tumor suppressor in this condition [38]. MICAL2 functions as a new gene that modulates EMT, a mechanism implicated in cancer proliferation and invasion. In pancreatic cancer, GO enrichment analysis indicated that MICAL2 is mostly associated with EMT, ECM architecture, and other biological activities pertinent to tumor metastasis [39].

In alignment with prior single-cell cancer research, ERG+ tumor cells formed distinct clusters per patient, apart from non-malignant epithelial clusters. Our examination of ERG-tumor cells revealed that these tumors did not cluster by patient, and we noted a common heterogeneity between ERG-tumor cells and non-malignant luminal cells [1,40]. The identification of tumor cells in this work mostly relied on copy number variation calculation, using non-malignant epithelial cells as a reference. Considering the limited sequencing depth of scRNA-seq, we recognize that minor localized CNV alterations may not have been adequately detected in our study [41].

Our study also highlights several previously underappreciated genes, such as TNC, THBS1, and PLOD2, as potentially critical ECM regulators of EMT. These genes were found to be co-expressed with canonical EMT markers and enriched in mesenchymal-like tumor regions. Notably, PLOD2, a lysyl hydroxylase involved in collagen cross-linking, has been recently implicated in ECM stiffening and metastasis across various cancers, including prostate cancer. These findings provide new candidate targets for therapeutic investigation. We found prostate cancer-enriched epithelial cell states in tumor tissues and in vitro organoid cultures from tumor specimens. Hillock cells, mesenchymal stem cells, and MKI67+ epithelial cells were found in organoid samples but not tumor tissues. Hillock cells multiply in organoid cultures but not in localized tumor tissue specimens. The mechanisms are unknown. Cell-state transitions in BE and club cells within organoids are better understood due to their proliferation. We found that prostate cancer epithelial organoids include numerous tissue-specific cell types and may be a good model for cell-state plasticity under selection pressures and genetic changes. We did not find a distinct NKX3-1+/KLK3+/AR+ luminal cell population in prostate organoids, unlike earlier studies. Our failure to culture differentiated luminal cells or identify them using single-cell sequencing may explain this. Most “tumor-like” cells were found in one patient’s organoid. After numerous passes, certain organoids may lose their “tumor-like” cells.

From a methodological standpoint, the combination of microarray and scRNA-seq data allowed for robust cross-validation of findings, leveraging the strengths of each platform. While microarrays provided broad coverage and statistical power for gene co-expression analysis, scRNA-seq enabled fine-grained dissection of cell-specific expression patterns and tumor heterogeneity. The convergence of findings across platforms reinforces the biological significance of our results. Despite these advances, several limitations should be noted. First, while transcriptomic data provide powerful insights into gene expression dynamics, they do not fully capture post-transcriptional modifications or protein activity. Future studies integrating proteomic, epigenomic, and metabolomic data would offer a more comprehensive view. Second, although we inferred cell–cell interactions computationally, experimental validation using co-culture systems or in vivo models would strengthen the causal interpretations of stromal–epithelial crosstalk in EMT.

Our integrated analysis reveals a complex regulatory network linking extracellular matrix (ECM) remodeling with epithelial–mesenchymal transition (EMT) in prostate cancer. The identification of gene co-expression modules containing classical EMT markers (e.g., VIM, ZEB1, SNAI2) alongside ECM-related genes (COL1A1, FN1, SPARC, LOX) underscores a tightly coordinated interaction driving tumor cell plasticity and invasion. This supports the hypothesis that ECM dynamics are not merely structural changes but active participants in promoting EMT and metastatic potential. Additionally, the discovery of a unique club epithelial cell population in prostate cancer tissues suggests previously unrecognized cellular heterogeneity and potential progenitor-like roles contributing to tumor progression. Moreover, highlighting specific genes, such as *ITGBL1*, *DSC3*, *ANGPT1*, *ARMCX1*, *MICAL2*, *TNC*, *THBS1*, and *PLOD2*, offers new insights into molecular mechanisms by which ECM remodeling may facilitate EMT, tumor proliferation, and metastasis. For instance, PLOD2’s role in collagen cross-linking and ECM stiffening has direct implications for the biomechanical environment favoring cancer cell dissemination.

Clinically, these findings provide promising candidate biomarkers and therapeutic targets. The consistent enrichment of ECM remodeling genes with EMT markers in aggressive prostate cancer phenotypes suggests these molecules could serve as prognostic indicators of metastatic risk. For example, the overexpression of PLOD2 and LOX has been correlated with poor outcomes in multiple cancers, and their detection in prostate tumors may help stratify patients who would benefit from more aggressive treatments or ECM-targeting therapies.

Furthermore, our identification of prostate cancer-specific epithelial cell states, including the novel club cell population and proliferative MKI67+ epithelial cells in organoid cultures, offers potential models for testing drug responses and understanding tumor heterogeneity in vitro. This could accelerate personalized medicine approaches by enabling functional studies on patient-derived cells that recapitulate key tumor features. In sum, our study provides a comprehensive transcriptomic framework that deepens the understanding of how ECM and EMT interplay promotes prostate cancer progression. It lays the groundwork for future experimental validation and clinical translation aimed at improving diagnostic precision and therapeutic interventions.

While our findings highlight the potential genetic co-evolution between tumor and stromal cells, we acknowledge that experimental validation using co-culture systems or in vivo models would further strengthen these insights. Due to financial limitations, we were unable to perform such laboratory-based experiments in the current study. However, it is worth noting that numerous recent studies have successfully relied solely on integrative bioinformatics approaches to generate biologically meaningful hypotheses, and our study similarly provides a comprehensive computational framework that can guide future experimental investigations.

## 4. Materials and Methods

### 4.1. Microarray Data

The gene expression dataset (GSE55945 [42], GSE104749 [43], GSE46602 [44], and GSE32571 [45]) was obtained from the Gene Expression Omnibus (GEO) website (https://www.ncbi.nlm.nih.gov/geo/, accessed on 12 August 2024) for the present investigation. The gene expression collection included a total of 96 samples, 36 of which were cancerous and 36 of which were normal, from the prostate. In Table 3 you can see a summary of all the databases. Evidence of gene expression in certain organs was validated by the cBioPortal study.

### 4.2. Data Processing

R, a statistical package available at https://www.r-project.org/ (accessed on 3 August 2024), was used to examine the expression data. In order to analyze the DEGs, we followed the selection criteria of a significant threshold of *p*  <  0.05 and an absolute t-value greater than 2. Additionally, for multiple testing correction, the Benjamini–Hochberg method was applied where necessary to control the false discovery rate (FDR). Our team obtained a comprehensive catalog of enzymes from the PANTHER that play a role in the interconversion of metabolites [46]. Furthermore, we used the Venn diagram tool inside the R programming environment (VennDiagram package) to identify the overlap of DEGs, oncogenes, and tumor suppressor genes (TSGs), considering only genes consistently found across datasets [46]. Furthermore, we used the Venn diagram tool inside the R programming environment to identify the overlap of DEGs, oncogenes, and tumor suppressor genes (TSGs).

### 4.3. Extracellular Matrix in Epithelial–Mesenchymal Transition Genes in PCa

To examine the simultaneous expression of CDC42, TAGLN, and GSN genes at the transcript level in normal prostate tissue, we used the “correlation analysis” tool available in the GEPIA2 [47] database (http://gepia2.cancer-pku.cn, accessed on 12 Augest 2024) and applied the Pearson correlation coefficient approach, considering |r| > 0.4 and *p*  <  0.01 as significant. For the co-expression analysis, we selected a curated panel of mesenchymal marker genes, including *VIM*, *SNAI1*, *ZEB1*, *TWIST1*, and *FN1*, which are typically upregulated in tumors and associated with EMT activation. These genes were chosen based on established literature and validated EMT signatures. Epithelial markers, such as CDH1 (E-cadherin), which are typically downregulated during EMT, were not included in the initial co-expression analysis but were considered during comparative expression profiling.

### 4.4. Transcription Factors Regulating EMT Genes

We extracted and evaluated data on the transcription factor and the control of these interactions from the TRUST database (https://www.grnpedia.org/trrust/, accessed on 11 August 2024). According to these findings, high-risk and low-risk polyps express these genes quite differently [48,49,50,51].

### 4.5. EMT Co-Expressed Genes and PPI Network

Samples from TCGA COAD tumors, TCGA neighboring normal tissues, and GTEx colon were analyzed using the GEPIA2 algorithm to find the 100 genes that were found to be co-expressed with each metabolite interconversion enzyme gene. Metabolite interconversion enzyme genes are part of the protein–protein interaction (PPI) network that was sourced from the STRING database (http://string-db.org, accessed on 4 August 2024). The interaction score has to meet a minimum requirement of 0.4. So, we got rid of nodes that didn’t have an edge or weren’t part of the main integrated network [2,3,33,52,53].

### 4.6. EMT Genes Functional Enrichment Analysis

In addition, the Enrichr database [54] (https://maayanlab.cloud/Enrichr/, accessed on 17 August 2024) was utilized to conduct Gene Ontology (GO) enrichment analysis and Kyoto Encyclopedia of Genes and Genomes (KEGG) enrichment analysis on the co-expressed genes mentioned earlier. The studies were carried out separately for the genes encoding metabolite interconversion enzymes in GTEx healthy colon, normal tissues around prostate cancer, and cancerous prostate tissues. As a result, we used the “ggplot” package in R to display the most important features according to their *p*-values, and we only included records with a *p*-value less than 0.05. The analysis was conducted on genes co-expressed with metabolite interconversion enzymes identified from prostate tumor and normal tissue datasets. Only terms with an adjusted *p*-value of less than 0.05 were considered statistically significant. The significance of enriched terms was calculated using Fisher’s exact test based on the hypergeometric distribution. To visualize the top enriched GO terms and KEGG pathways, we used the “ggplot2” package in R, selecting the most relevant features based on their *p*-values. This approach allowed us to identify key biological processes and pathways, such as extracellular matrix organization, immune system regulation, and cell adhesion, that are potentially involved in EMT during prostate cancer progression.

### 4.7. Exploring GO and KEGG Pathway

Ontology terms (Biological Process (BP), Cellular Component (CC), and Molecular Function (MF)) and pathways related to colorectal cancer (CRC) ontology terms, including KGs, were analyzed using GO functions and KEGG pathway enrichment. In order to assess the significantly enhanced GO terms and KEGG pathways in the cDEGs, including KGs, we label Si as the collection of annotated genes linked to a certain biological process in the database. Where i ranges from 1 to r, Mi is the number of genes in Si. The whole combined set, which is the complement of Si, consists of all annotated genes, and N is the total number of these genes. A total of n DEGs should be considered, where ki is the number of DEGs that match the annotated gene set. The contingency table reveals the issue. We computed the *p*-value using Fisher’s exact test statistic, which is based on the hypergeometric distribution, in order to find the highly enriched GO keywords and KEGG pathways that are connected with our suggested cDEGs. To perform Fisher’s exact test, we turned to the web-based Enrichr tool.

### 4.8. Regulatory Network Analysis of KGs

Using the publicly available JASPAR database, we constructed a network of interactions between TFs and KGs to identify the main transcription factors (TFs) that control KGs. The NetworkAnalyst was used to create the interaction network. Using the open-source TarBase v8.0 (Release7.0) web tool, we built an interaction network between KGs and miRNAs to find the main miRNAs that control KGs after transcription. From the networks, the most notable miRNAs (miRNAs-KGs) were chosen and identified as essential miRNAs.

### 4.9. WGCNA

If the mean FPKM was more than 0.5, it was discarded. In addition, we transform the similarity matrix into an adjacency matrix by doing cluster analysis on each sample and fitting index analysis. We conclude by finding the suitable power (soft threshold) value to ensure a correlation of more than 0.9 between connectivity and power. It was determined that 14 was the ideal power value. We used this information to build a topology overlap matrix (TOM) and a network that does not scale. We also developed a gene (tree graph) hierarchical clustering tree for module detection utilizing the function, which was based on the corresponding TOM dissimilarity (diss TOM). The necessary parameters were reduced in order to avoid the mass production of modules. With MEDissThres = 0.25, Module Size = 30, and Deep Split = 2, we obtain a similarity value of 0.75. A new merging module is created when the similarity between the modules is greater than 0.75 [55].

### 4.10. Data Collection and Preprocessing of Single Cells

Patients with prostate cancer who had neoadjuvant treatment had their transcriptionome and clinical data pulled from the GEO database. Using the following identifiers: GSE94577, GSE82225, and GSE176031, the research was retrieved from https://www.ncbi.nlm.nih.gov/geo/ (accessed on 17 August 2024). The discovery cohort was the biggest and most thorough collection of breast cancer samples ever compiled, with a total of 306 samples. In the two distinct validation cohorts, twenty breast cancer samples were taken from GSE82225 and fifteen from GSE94577. Specimens without complete survival data were omitted from the research [56].

### 4.11. EMT Genes Expression and Survival Analysis in PCa

Utilizing the GEPIA2 web-based tool (http://gepia2.cancer-pku.cn, accessed on 17 August 2024), we investigated the mRNA expression levels of metabolite interconversion enzymes. As part of the expression investigation, we compared PCa data from TCGA with healthy samples from GTEx, a database that stores genotype-tissue expression information. In addition, we evaluated the association between PCa survival and genes encoding metabolite interconversion enzymes using GEPIA2. The clinical data and gene expression data collected from TCGA were both used in this investigation. At a *p*-value of 0.05, statistical significance was determined.

### 4.12. Examining Gene Involved in EMT in Relation to Immunological Subtypes

The correlation between EMT levels and six immunological subtypes of prostate cancer was examined in this research. The research made use of TISIDB, which can be accessed at http://cis.hku.hk/TISIDB/index.php (accessed on 15 August 2024), an integrated repository portal for studying interactions between the tumor-immune system. Readers interested in learning more about the molecular features of tumor-immune interactions in different forms of cancer are referred to the Tumor Immune Estimation Resource (TIMER) at https://cistrome.shinyapps.io/timer/ (accessed on 15 August 2024). TIMER uses tumor-infiltrating immune cell types (B cells, CD4+ T cells, CD8+ T cells, macrophages, neutrophils, and dendritic cells, or DCs) and data from The Cancer Genome Atlas (TCGA) to calculate their relative abundance using a deconvolution statistical approach.

### 4.13. Expression Analysis for KGs by GEPIA

To evaluate the expression levels of certain genes, we used a tool known as gene expression profiling interactive analysis (GEPIA, GSE137829 [57]). This program enabled us to analyze data from the TCGA databases and compare the expression levels of significant genes in prostate tissues with those in normal tissues [9,32,33,49,50,58,59].

## 5. Conclusions

In conclusion, our integrative transcriptomic analysis reveals that ECM remodeling is both a marker and driver of EMT in prostate cancer, facilitated by tumor–stromal interactions and spatially organized within the tumor microenvironment. These insights have important implications for understanding prostate cancer progression and for the development of therapeutic strategies targeting ECM-mediated pathways. Future directions may include the application of spatial transcriptomics and functional perturbation studies to validate the molecular interactions identified here and explore their therapeutic potential.

## Figures and Tables

**Figure 1 ijms-26-08575-f001:**
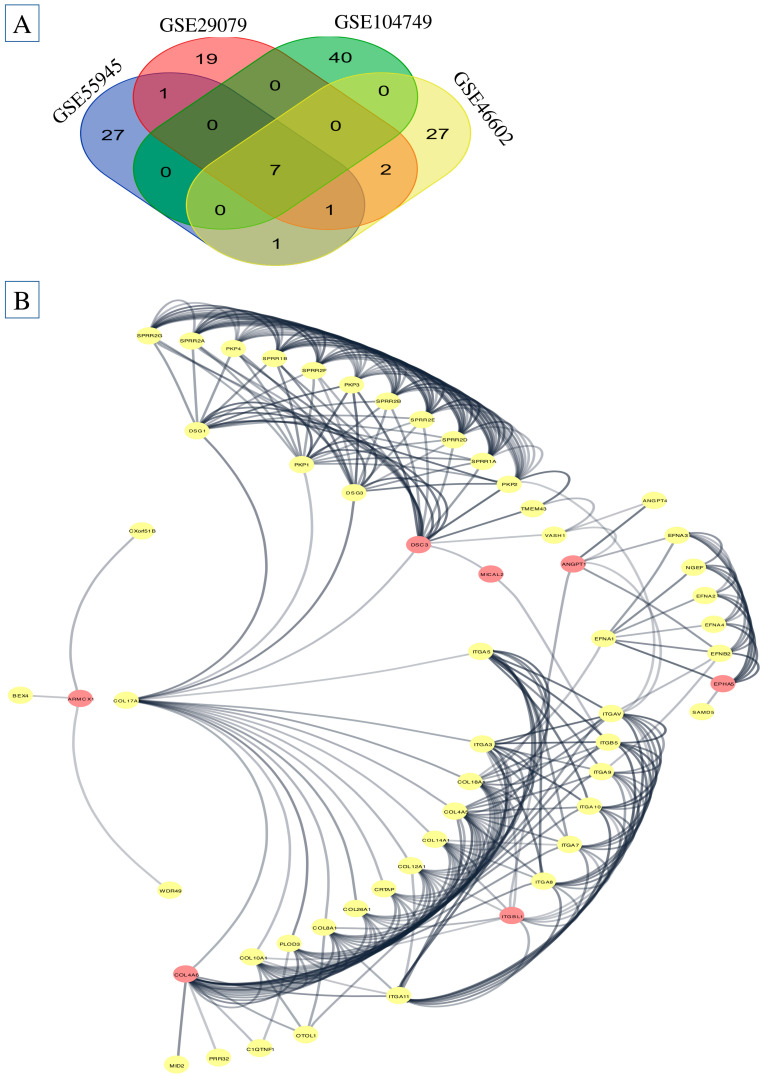
Using Enrich, we analyze PPI. (**A**) The study includes four datasets: GSE55945, GSE29079, GSE104749 and GSE46602. (**B**) PPI is involved in EMT in PCa.

**Figure 2 ijms-26-08575-f002:**
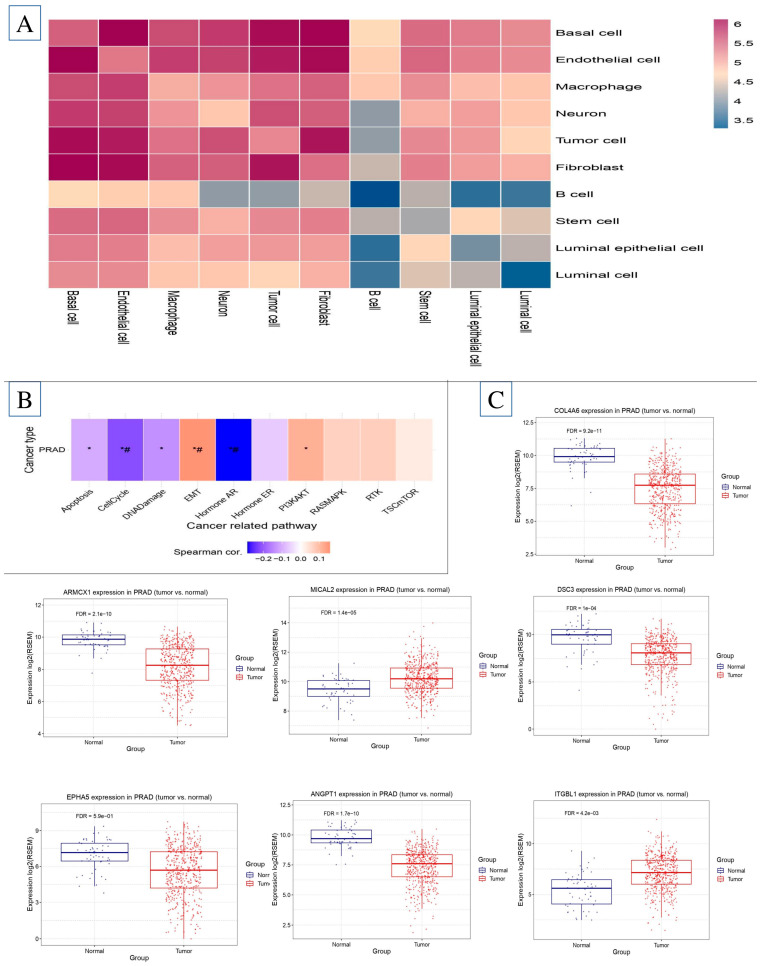
Correlation between cell–cell gene expression and gene expression analysis. (**A**) Correlation between cell–cell gene expression in prostate cancer; (**B**) collect gene involved in EMT gene in PCa; (**C**) identify EMT genes present in the microenvironment of prostate cancer and determine the levels of mRNA expression for the genes *ARMCX1*, *BEX4*, *TMEM43*, *VASH1*, *C1QTNF1*, *COL10A1*, and *COL4A6* in tissue samples of prostate cancer using the GSE database. * means *p*-value < 0.05, and # means *p*-value < 0.001.

**Figure 3 ijms-26-08575-f003:**
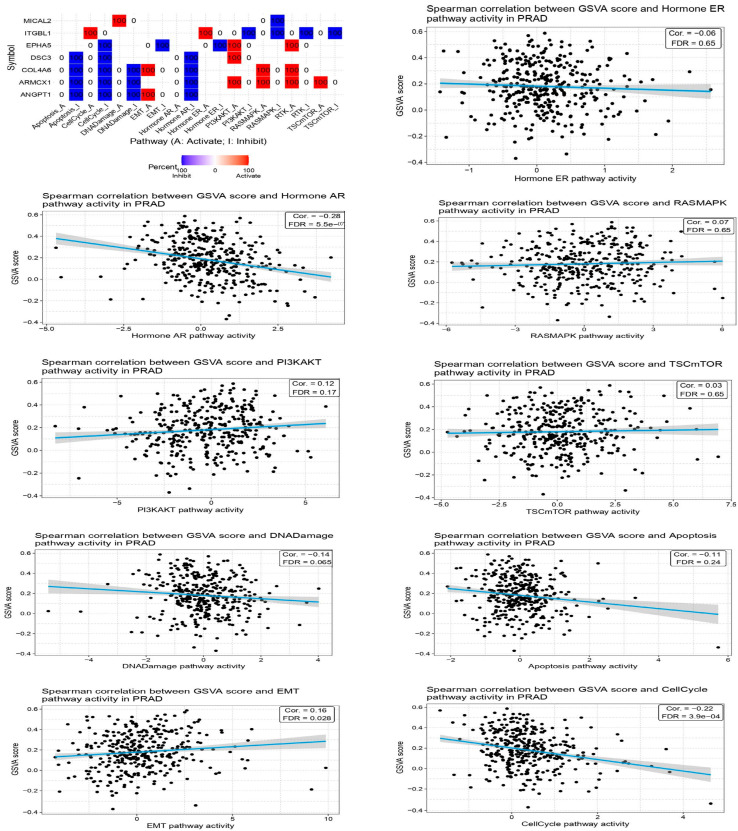
Identification of the signaling pathway linked to EMT in the prostate cancer microenvironment. The signaling pathways involved in the prostate cancer environment include the Hormone ER pathway, Hormone AR pathway, RAS-MAPK pathway, TSC-mTOR pathway, DNA damage pathway, and cell cycle pathway.

**Figure 4 ijms-26-08575-f004:**
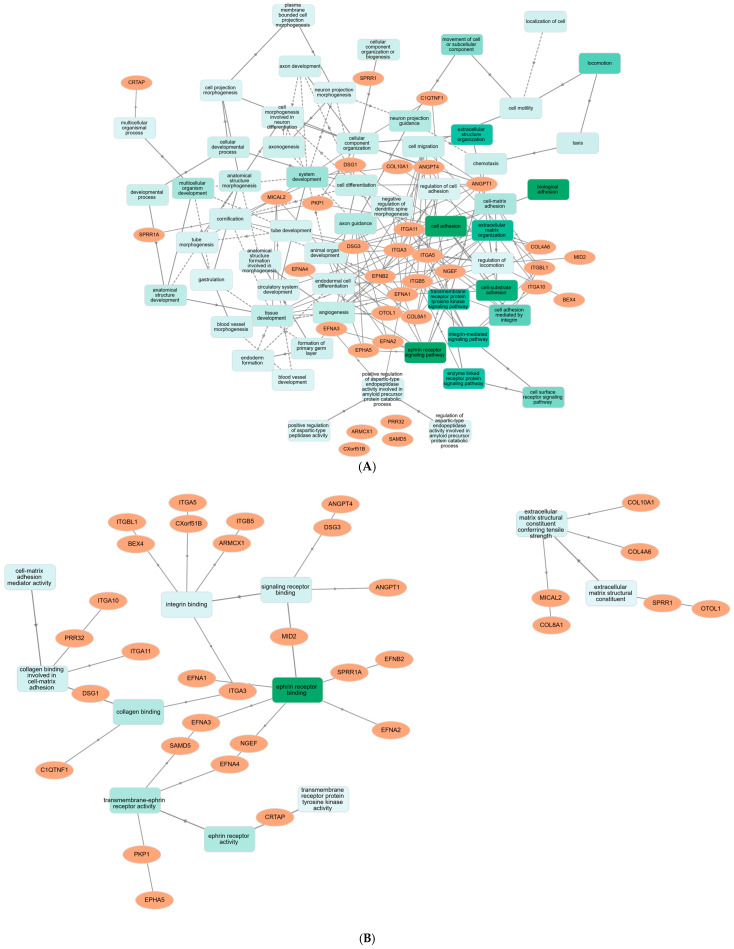
Biological processes and molecular functions in the genes. (**A**) analysis of biological processes of PPI, (**B**) analysis of molecular function of PPI.

**Figure 5 ijms-26-08575-f005:**
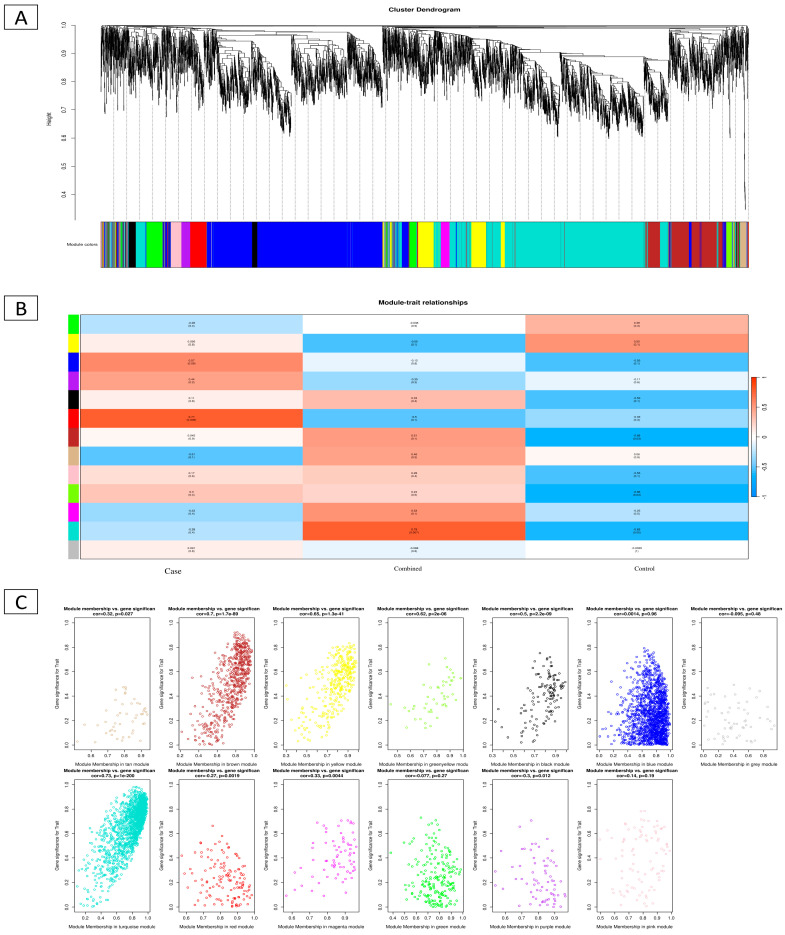
Identify and sort a hub gene module that exhibits a correlation in PCa. (**A**) Identifying the correlation between PPI, WGCNA, and GO enrichment and branches of the cluster dendrogram of the most connected genes gave rise to 7 gene co-expression modules; (**B**) heat map of the correlation between module 13 and phenotype; and (**C**) intergenic connectivity of prostate cancer cells’ genes in the turquoise module.

**Figure 6 ijms-26-08575-f006:**
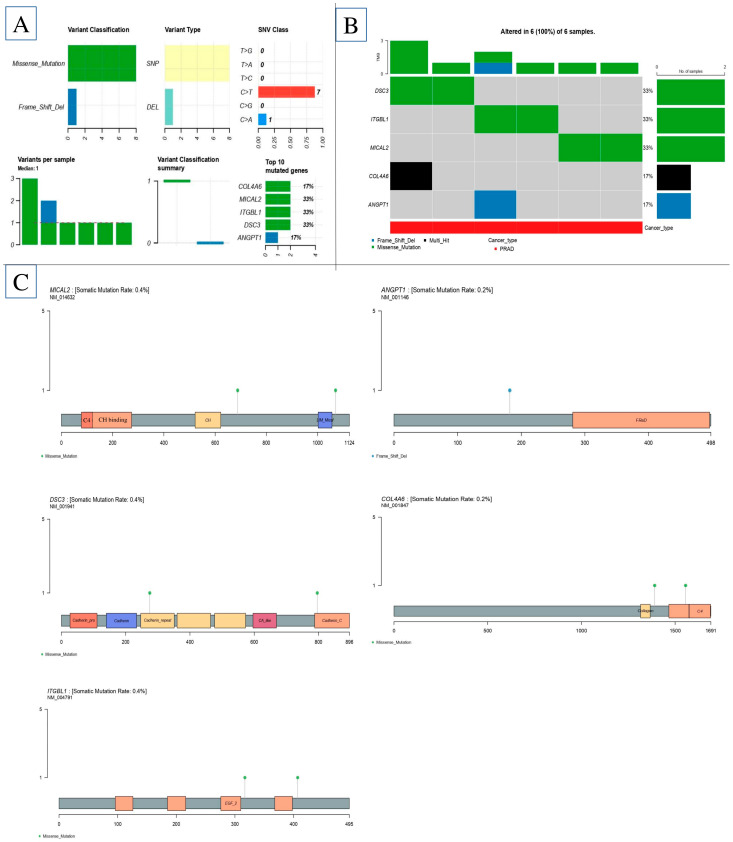
A regional association map and the survival probability of both high- and low-risk categories are shown for the gene area encompassing the *ARMCX1*, *BEX4*, *TMEM43*, *VASH1*, *C1QTNF1*, *COL10A1*, and *COL4A6* genes. Several prostate cancer-associated SNPs in this area exhibit varied amounts of linkage disequilibrium (LD). (**A**) The examined gene is below the SNPs’ chromosomal positions on the *x*-axis. Regressing normalized gene expression levels on the number of minor alleles of each SNP genotype, adjusted for histologic features and 14 expression main components, yields the −log10(*p* value) on the *y*-axis. (**B**) A diamond represents the eQTL result and a dotted red vertical line represents the PrCa-risk SNP. A regional association map for the (**C**) genes (*ARMCX1*, *BEX4*, *TMEM43*, *VASH1*, *C1QTNF1*, *COL10A1*, and *COL4A6*) area shows multiple prostate cancer risk-associated SNPs with varied degrees of linkage disequilibrium (LD).

**Figure 7 ijms-26-08575-f007:**
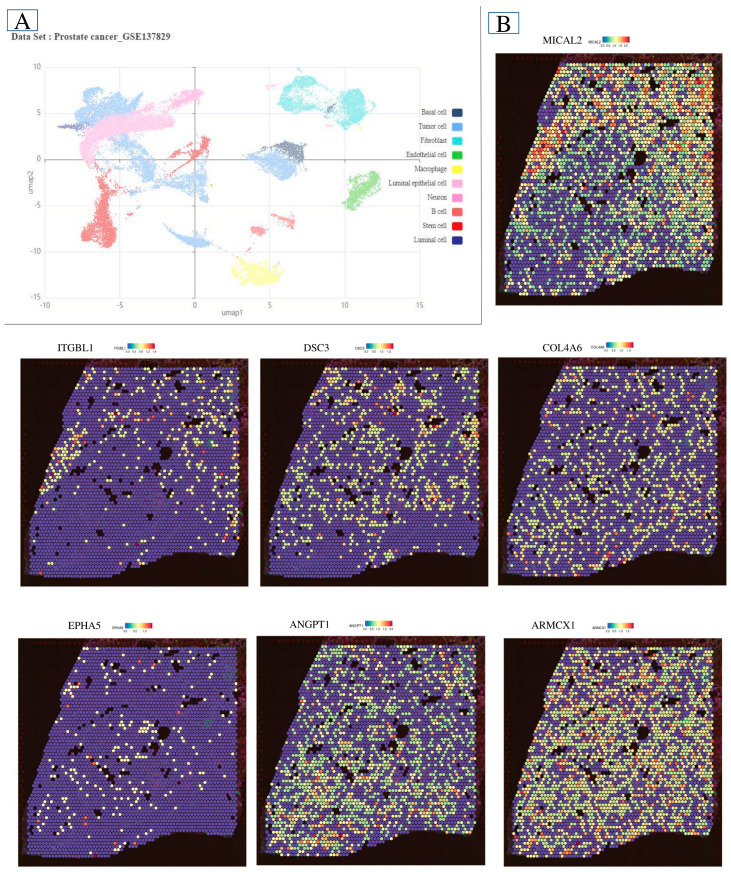
Characteristics of PCa microenvironment in single-cell RNA sequencing. (**A**) Ten clusters were found in 6 normal and 6 PCa samples, demonstrating the prostate’s cell variety. (**B**) Changes in *ARMCX1*, *BEX4*, *TMEM43*, *VASH1*, *C1QTNF1*, *COL10A1*, and *COL4A6* imply prostate cancer etiology involvement.

**Figure 8 ijms-26-08575-f008:**
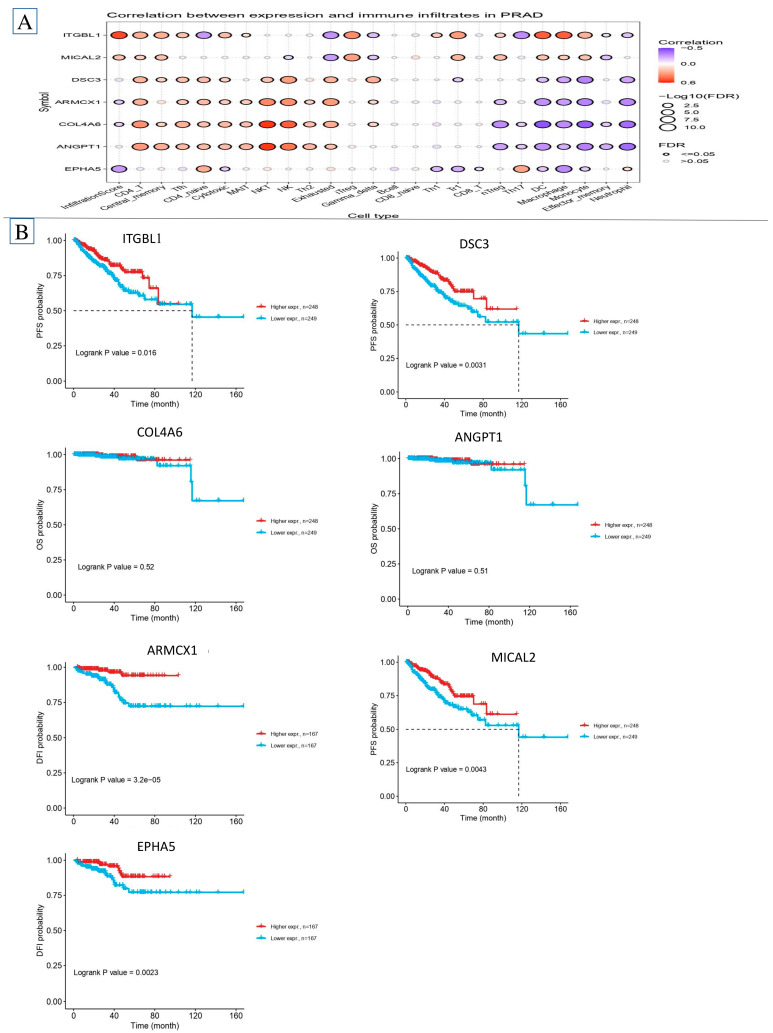
Identify immune cells present in (**A**) determine the levels of mRNA expression and (**B**) their survival plot using the GSE database.

**Figure 9 ijms-26-08575-f009:**
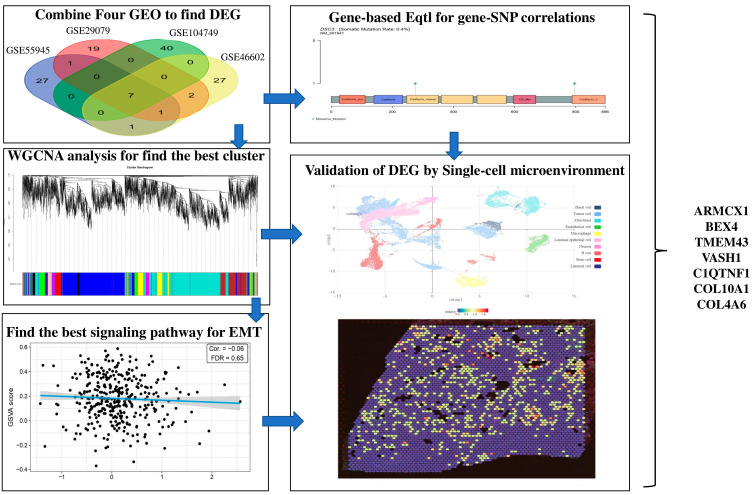
Flowchart illustrating the integrative bioinformatics pipeline used to identify and validate EMT-related genes in prostate cancer. The workflow includes DEG screening, PPI network construction, functional enrichment analysis, WGCNA, single-cell expression profiling, and clinical validation through survival and eQTL analyses.

**Table 1 ijms-26-08575-t001:** Summary of GEO datasets used in the study.

Dataset	GEO Accession	Platform	Total Samples	Cancer Samples	Normal Samples	Assigned Subtypes	Clinical Metadata
GSE55945	GPL570	Affymetrix Human Genome U133 Plus 2.0 Array	72	36	36	Luminal A (12), Luminal B (14), Basal (10)	Limited (age, diagnosis)
GSE104749	GPL10558	Illumina HumanHT-12 V4.0 Expression BeadChip	20	16	4	Luminal B (6), Basal (5), Neuroendocrine (5)	Gleason score provided
GSE46602	GPL570	Affymetrix Human Genome U133 Plus 2.0 Array	50	36	14	Luminal A (10), Basal (15), Neuroendocrine (11)	Gleason score, TNM stage
GSE32571	GPL6244	Affymetrix Human Gene 1.0 ST Array	22	13	9	Luminal B (4), Basal (5), Neuroendocrine (4)	Very limited

**Table 2 ijms-26-08575-t002:** Differential expression of key EMT genes across PCa datasets.

Gene	log_2_FC (GSE55945)	log_2_FC (GSE104749)	log_2_FC (GSE46602)	log_2_FC (GSE32571)	FDR (Avg)	Subtype(s) Highly Expressed
*ARMCX1*	+2.4	+1.9	+2.7	+2.1	<0.001	Luminal A, Basal
*BEX4*	+1.8	+2.0	+2.5	+2.2	<0.001	Basal
*TMEM43*	−2.1	−2.3	−1.9	−2.0	<0.001	Downregulated in all
*VASH1*	+1.6	+1.7	+2.1	+1.9	0.002	Luminal B
*C1QTNF1*	+2.2	+2.5	+2.8	+2.6	<0.001	Neuroendocrine
*COL10A1*	+2.9	+3.0	+3.4	+3.1	<0.001	All subtypes
*COL4A6*	−2.5	−2.1	−2.7	−2.3	<0.001	Downregulated in all

**Table 3 ijms-26-08575-t003:** We are studying the GEO related to PCa using mRNA and Single-cell RNA sequencing.

Sample/Accession No.	Prostate Cancer	Benign Prostate Hyperplasia	Sample Type	Platform	RNA Type
GSE55945	8	8	tissue	Affymetrix GPL570	mRNA
GSE104749	4	4	tissue	Affymetrix GPL570	mRNA
GSE46602	36	14	tissue	Affymetrix GPL570	mRNA
GSE32571	59	39	tissue	Affymetrix GPL6947	mRNA
GSE137829	6	6	Tissue	GPL24676 Illumina NovaSeq 6000	mRNA/Single cell

## Data Availability

The data that support the findings of this study are available from the corresponding author upon reasonable request or visit https://zenodo.org/records/15586926 (accessed on 12 August 2024).

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
