# Peer review of "Analysis of Microarray and Single-Cell RNA-Seq Finds Gene Co-Expression and Tumor Environment Associated with Extracellular Matrix in Epithelial–Mesenchymal Transition in Prostate Cancer"

_ijms, 2025, doi:10.3390/ijms26178575_

Round 1

Reviewer 1 Report (New Reviewer)

Comments and Suggestions for Authors

The paper entitled “Analysis of microarray and single-cell RNA-seq finds gene coexpression, and tumor environment associated with Extracellular Matrix in Epithelial-Mesenchymal Transition in prostate cancer” that you kindly submitted for publication in the “International Journal of Molecular Sciences” now been considered. 

In this manuscript, the authors analyzed several microarray datasets derived from prostate cancer tissues and identified an interesting gene-gene network. Furthermore, differentially expressed genes (DEGs) were examined in the context of epithelial-to-mesenchymal transition (EMT), which was further explored through single-cell RNA sequencing of prostate cancer tissue specimens. This approach led to the identification of targets associated with the tumor microenvironment during prostate cancer progression. While the authors' methods are interesting and generally well-conceived, the manuscript requires improvements before it can be considered for publication. 

The analytical strategy should be explained more comprehensively throughout the manuscript. Including a flowchart to illustrate the overall study design from DEGs upto seven genes would greatly enhance clarity.  

Figures shown in the manuscript should be exchanged to better resolution (at this stage, main findings are not clearly expressed).

Additionally, the authors should elaborate on the biological and clinical significance of their results to help readers better understand their implications.

Comments on the Quality of English Language

The English could be improved to more clearly express the research.

Author Response

Dear Editor and Reviewers,

Thanks for taking the time to carefully review our manuscript entitled “Analysis of microarray and single-cell RNA-seq finds gene co-expression, and tumor environment associated with Extracellular Matrix in Epithelial-Mesenchymal Transition in prostate cancer.”, The comments are illuminating for our present and further work.

Sincerely,

Reviewer 1

The paper entitled “Analysis of microarray and single-cell RNA-seq finds gene coexpression, and tumor environment associated with Extracellular Matrix in Epithelial-Mesenchymal Transition in prostate cancer” that you kindly submitted for publication in the “International Journal of Molecular Sciences” now been considered. 

In this manuscript, the authors analyzed several microarray datasets derived from prostate cancer tissues and identified an interesting gene-gene network. Furthermore, differentially expressed genes (DEGs) were examined in the context of epithelial-to-mesenchymal transition (EMT), which was further explored through single-cell RNA sequencing of prostate cancer tissue specimens. This approach led to the identification of targets associated with the tumor microenvironment during prostate cancer progression. While the authors' methods are interesting and generally well-conceived, the manuscript requires improvements before it can be considered for publication. 

The analytical strategy should be explained more comprehensively throughout the manuscript. Including a flowchart to illustrate the overall study design from DEGs upto seven genes would greatly enhance clarity.  

Reply: Done “Flowchart illustrating the integrative bioinformatics pipeline used to identify and validate EMT-related genes in prostate cancer. The workflow includes DEG screening, PPI network construction, functional enrichment analysis, WGCNA, single-cell expression profiling, and clinical validation through survival and eQTL analyses.”.

Figures shown in the manuscript should be exchanged to better resolution (at this stage, main findings are not clearly expressed).

Reply: Done. We replace with high resolustion.

Additionally, the authors should elaborate on the biological and clinical significance of their results to help readers better understand their implications.

Reply: Thank you for your valuable suggestion to elaborate on the biological and clinical significance of our findings. We have expanded the Discussion section to more clearly highlight the potential implications of our results in both biological understanding and clinical practice.

Reviewer 2 Report (New Reviewer)

Comments and Suggestions for Authors

In the present study, the authors aimed to identify the correlation between the tumor extracellular matrix and EMT in prostate cancer (PCa). They employed several gene expression datasets from the GEO database to explore the profile of EMT-associated extracellular matrix genes. Their findings may be valuable for researchers investigating the pro-tumor effects of the microenvironment in cancer biology. However, this manuscript contains several flaws that need to be addressed. Below are some suggestions for the authors.

Major concerns

  1. The authors used several bioinformatic tools to identify significant genes associated with EMT; however, they did not provide detailed criteria for how each tool was used to select these genes. I suggest that the authors improve the Materials and Methods section by including this information.

2. The authors identified several genes that were co-expressed with EMT; however, it is unclear which EMT-related genes were used for the co-expression analysis. Did the authors select mesenchymal-related genes (typically upregulated in tumors) or epithelial-related genes (typically downregulated in tumors)?

3. The authors only presented the patients' survival rates based on gene expression levels analyzed by multiple bioinformatic tools. However, they did not provide the expression levels of each gene in tumor versus normal tissues. In my opinion, the clinical significance of some genes may differ between their impact on survival and their expression patterns in tumors. The authors should also report the expression levels of ITGBL1, DSC3, COL4A6, ANGPT1, ARMCX1, MICAL2, and EPHA5 in tumor and normal tissues, rather than focusing solely on the clinical relevance of these candidate genes based on patient outcomes.

4. The authors should describe whether the genes mentioned above have been previously reported to regulate EMT. This would enhance the credibility of the genes identified using the authors' strategy.

Author Response

Dear Editor and Reviewers,

Thanks for taking the time to carefully review our manuscript entitled “Analysis of microarray and single-cell RNA-seq finds gene co-expression, and tumor environment associated with Extracellular Matrix in Epithelial-Mesenchymal Transition in prostate cancer.”, The comments are illuminating for our present and further work.

Sincerely,

Reviewer 2

In the present study, the authors aimed to identify the correlation between the tumor extracellular matrix and EMT in prostate cancer (PCa). They employed several gene expression datasets from the GEO database to explore the profile of EMT-associated extracellular matrix genes. Their findings may be valuable for researchers investigating the pro-tumor effects of the microenvironment in cancer biology. However, this manuscript contains several flaws that need to be addressed. Below are some suggestions for the authors.

Major concerns

  1. The authors used several bioinformatic tools to identify significant genes associated with EMT; however, they did not provide detailed criteria for how each tool was used to select these genes. I suggest that the authors improve the Materials and Methods section by including this information.

 Reply: We rewrite and improve the material and methods (Lines 490-497).

  1. The authors identified several genes that were co-expressed with EMT; however, it is unclear which EMT-related genes were used for the co-expression analysis. Did the authors select mesenchymal-related genes (typically upregulated in tumors) or epithelial-related genes (typically downregulated in tumors)?

  Reply: For the co-expression analysis, we selected a curated panel of mesenchymal marker genes, including VIM, SNAI1, ZEB1, TWIST1, and FN1, which are typically upregulated in tumors and associated with EMT activation. These genes were chosen based on established literature and validated EMT signatures. Epithelial markers such as CDH1 (E-cadherin), which are typically downregulated during EMT, were not included in the initial co-expression analysis but were considered during comparative expression profiling (Lines 508-513).

  1. The authors only presented the patients' survival rates based on gene expression levels analyzed by multiple bioinformatic tools. However, they did not provide the expression levels of each gene in tumor versus normal tissues. In my opinion, the clinical significance of some genes may differ between their impact on survival and their expression patterns in tumors. The authors should also report the expression levels of ITGBL1, DSC3, COL4A6, ANGPT1, ARMCX1, MICAL2, and EPHA5 in tumor and normal tissues, rather than focusing solely on the clinical relevance of these candidate genes based on patient outcomes.

  Reply: Thank you for this valuable feedback. We acknowledge the importance of presenting gene expression levels in tumor versus normal tissues alongside survival analysis to fully understand the clinical significance of these candidate genes.

In response, we would like to highlight that Table 2 and Figure 2B already provide detailed data on the differential expression of key genes, including ITGBL1, DSC3, COL4A6, ANGPT1, ARMCX1, MICAL2, and EPHA5, across multiple prostate cancer datasets. Specifically:

Table 2 summarizes the logâ‚‚ fold changes and adjusted p-values (FDR) for these genes, showing significant upregulation or downregulation in tumor tissues relative to normal controls across different GEO datasets.

Figure 2B visually represents the involvement of these genes in the epithelial-mesenchymal transition (EMT) process within the prostate cancer microenvironment, supporting their differential expression and functional roles.

Together, these data demonstrate the expression patterns of the key genes in tumors compared to normal tissue, complementing the survival analyses presented in the manuscript. We agree that integrating these expression profiles with clinical outcome data offers a more comprehensive understanding of their roles in prostate cancer progression.

  1. The authors should describe whether the genes mentioned above have been previously reported to regulate EMT. This would enhance the credibility of the genes identified using the authors' strategy.

 Reply: Several of the genes highlighted in our study—ITGBL1, DSC3, COL4A6, ANGPT1, ARMCX1, MICAL2, and EPHA5—have been previously reported in the literature to play roles in EMT regulation or related processes:

ITGBL1: Known to modulate EMT by influencing cell adhesion and extracellular matrix interactions, promoting cancer cell migration and invasion in multiple cancers.

DSC3 (Desmocollin-3): Functions as a component of desmosomes and its altered expression has been linked to changes in cell-cell adhesion during EMT, often acting as a suppressor of EMT.

COL4A6: A collagen family member involved in extracellular matrix remodeling; alterations in collagen expression can facilitate EMT by modulating the tumor microenvironment.

ANGPT1 (Angiopoietin-1): Plays a role in angiogenesis and vascular remodeling, with indirect links to EMT via effects on tumor stroma and cell signaling pathways.

ARMCX1: Though less studied in EMT, ARMCX1 has been implicated in mitochondrial dynamics and may influence EMT-related metabolic reprogramming.

MICAL2: Known to regulate actin cytoskeleton remodeling, a key component of EMT, promoting cancer cell motility and invasiveness.

EPHA5: A receptor tyrosine kinase family member involved in cell-cell communication; Eph receptors have been widely implicated in EMT regulation through signaling that affects cell adhesion and migration.

Round 2

Reviewer 2 Report (New Reviewer)

Comments and Suggestions for Authors

The authors had provided appropriate responses for all questions. I agree that this manuscript meets the criteria for publication.

This manuscript is a resubmission of an earlier submission. The following is a list of the peer review reports and author responses from that submission.

Round 1

Reviewer 1 Report

Comments and Suggestions for Authors

The Abroudi and collaborators study shows interesting insights on the network of differentially expressed genes during EMT of PCa using the analyses of bulk and single-cells (one study) derived from prostate cancer tumor tissues. The datasets used were from 5 studies in the GEO database. The combined bioinformatics analyses show major signaling pathways involved in the prostate cancer environment including Hormone ER and AR pathways, RAS-MAPK pathway, TSC-mTOR pathway, DNA damage pathway, and cell cycle pathway, which are commonly observed in PCa patient ‘studies. They suggest that the genes ITGBL1, DSC3, COL4A6, ANGPT1, ARMCX1, MICAL2, and EPHA5 as putative biomarker genes as regulatory arm between EMT and ECM remodeling EMT in PCa. There is no doubt that tumor associated stromal and immune cells support overall tumor growth and metastasis. The use of multiple specific-cell-type markers (associated with gene expression) has shown evidence and clarified multiple paracrine factors involved in interaction of stromal and the cells in tumor tissue. However, as the authors concluded, the mathematical probability of genetic co-evolution of tumor and stromal cells need to go through experimental validation using co-culture systems or in vivo models. This study provides some insights that it will be possible to find out the major factors/genes involved in the biological and physiological connections and therapeutically translation.

Comments on the Quality of English Language

No comments

Author Response

Dear Editor and Reviewers,

Thanks for taking the time to carefully review our manuscript entitled “Analysis of microarray and single-cell RNA-seq finds gene co-expression, and tumor environment associated with Extracellu-lar Matrix in Epithelial-Mesenchymal Transition in prostate cancer”, the comments are illuminating for our present and further work.

Now we have revised according to your comments, and the list of changes or a rebuttal against each individual point is being raised as follows:

Sincerely,

Hossein Azizi (Corresponding author), H.azizi@ausmt.ac.ir

Reviewer’s comments:

The Abroudi and collaborators study shows interesting insights on the network of differentially expressed genes during EMT of PCa using the analyses of bulk and single-cells (one study) derived from prostate cancer tumor tissues. The datasets used were from 5 studies in the GEO database. The combined bioinformatics analyses show major signaling pathways involved in the prostate cancer environment including Hormone ER and AR pathways, RAS-MAPK pathway, TSC-mTOR pathway, DNA damage pathway, and cell cycle pathway, which are commonly observed in PCa patient ‘studies. They suggest that the genes ITGBL1, DSC3, COL4A6, ANGPT1, ARMCX1, MICAL2, and EPHA5 as putative biomarker genes as regulatory arm between EMT and ECM remodeling EMT in PCa. There is no doubt that tumor associated stromal and immune cells support overall tumor growth and metastasis. The use of multiple specific-cell-type markers (associated with gene expression) has shown evidence and clarified multiple paracrine factors involved in interaction of stromal and the cells in tumor tissue.

However, as the authors concluded, the mathematical probability of genetic co-evolution of tumor and stromal cells need to go through experimental validation using co-culture systems or in vivo models. This study provides some insights that it will be possible to find out the major factors/genes involved in the biological and physiological connections and therapeutically translation.

Reply: We sincerely appreciate the reviewer’s insightful comment. We fully acknowledge the importance of validating our findings through experimental approaches such as co-culture systems or in vivo models. However, due to current financial constraints, we were unable to carry out such validation in this study.

That said, it is important to note that a substantial number of high-impact studies in recent years have relied entirely on comprehensive bioinformatics analyses to generate hypotheses and identify key molecular players. Our study similarly contributes to this growing body of work, offering a robust and data-driven foundation that can guide future experimental research. We have added this clarification, along with the acknowledgment of this limitation, in the revised Discussion section of the manuscript (lines 411-416). We attached the recent full bioinformatics analysis study:

  • Bischoff, P., Trinks, A., Obermayer, B. et al. Single-cell RNA sequencing reveals distinct tumor microenvironmental patterns in lung adenocarcinoma. Oncogene 40, 6748–6758 (2021). https://doi.org/10.1038/s41388-021-02054-3
  • Li, J., Long, S., Zhang, Y. et al. Single-cell transcriptome sequencing reveals new epithelial-stromal associated mesenchymal-like subsets in recurrent gliomas. acta neuropathol commun 13, 127 (2025). https://doi.org/10.1186/s40478-025-02036-6
  • Single-cell analysis of matrisome-related genes in breast invasive carcinoma: new avenues for molecular subtyping and risk estimation (https://www.frontiersin.org/journals/immunology/articles/10.3389/fimmu.2024.1466762/full).

Reviewer 2 Report

Comments and Suggestions for Authors
  1. in figure1 paragraph, please provide the full name for DEGs and give more details for EdgeR algorithms in the methods.
  2. For Figure 1A, please provide a clear and normal image without stretching. 
  3. Figure 1A GSE number does not match the figure legend
  4. in the first paragraph of results, 544 genes did not show in streched figure 1A.
  5. The inconsistency among results paragraph, figure1A and figure legend raises significant problem for this paper's core hypothesis of the study.

Author Response

Dear Editor and Reviewers,

Thanks for taking the time to carefully review our manuscript entitled “Analysis of microarray and single-cell RNA-seq finds gene co-expression, and tumor environment associated with Extracellular Matrix in Epithelial-Mesenchymal Transition in prostate cancer”, the comments are illuminating for our present and further work.

Now we have revised according to your comments, and the list of changes or a rebuttal against each individual point is being raised as follows:

Sincerely,

Hossein Azizi (Corresponding author), H.azizi@ausmt.ac.ir

Reviewer’s comments:

  1. in figure1 paragraph, please provide the full name for DEGs and give more details for EdgeR algorithms in the methods.

Reply: We sincerely appreciate the reviewer’s insightful comment (155-156).

  1. For Figure 1A, please provide a clear and normal image without stretching. 

Reply: Done.

  1. Figure 1A GSE number does not match the figure legend

Reply: We corrected them.

  1. in the first paragraph of results, 544 genes did not show in streched figure 1A.

Reply: Thank you for your observation. We apologize for the confusion. The actual number of differentially expressed genes (DEGs) identified was 54, not 544, and we have corrected the text in the Results section accordingly. Figure 1A accurately reflects the 54 DEGs that met our statistical threshold (P < 0.05 and |t-value| > 2). We appreciate the reviewer’s attention to this detail.

  1. The inconsistency among results paragraph, figure1A and figure legend raises significant problem for this paper's core hypothesis of the study.

Reply: We acknowledge the earlier inconsistency between the reported number of DEGs in the results paragraph, Figure 1A, and the corresponding figure legend. This was due to a typographical error in the manuscript text, where “544” was mistakenly written instead of “54.” We have corrected this to maintain consistency across the results and ensure accurate representation of the core findings. Importantly, this correction does not affect the integrity of the downstream analyses or the overall conclusions of the study.

Round 2

Reviewer 2 Report

Comments and Suggestions for Authors
  1. In line 123 of the introduction, please give the full name of NEPC.
  2. In the results part, please provide more details for each PCa subtype. For example, what is the subtype for GSE55945? 
  3. The number of DEGs is 544, but the total number in the pie chart is 125 genes. Please explain more about it.
  4. For Figure 1B, please provide a clear figure; the current figure is unclear. Please 
  5. For Figure 1B, please give more details for each important gene, such as the log2 fold change value. A heatmap is a better way to show the differential expression.
  6. in line 182, there is no figure 1c and d. Please correct this.
  7. Figures 2 and 3 are unclear to see; please provide a clear version. 
  8.  the order of 6, 7,8 is not correct.